# Fish Collagen Peptides Protect against Cisplatin-Induced Cytotoxicity and Oxidative Injury by Inhibiting MAPK Signaling Pathways in Mouse Thymic Epithelial Cells

**DOI:** 10.3390/md20040232

**Published:** 2022-03-28

**Authors:** Won Hoon Song, Hye-Yoon Kim, Ye Seon Lim, Seon Yeong Hwang, Changyong Lee, Do Young Lee, Yuseok Moon, Yong Jung Song, Sik Yoon

**Affiliations:** 1Department of Urology, Pusan National University Yangsan Hospital, Yangsan 626-870, Korea; luchen99@hanmail.net; 2Immune Reconstitution Research Center of Medical Research Institute, Pusan National University College of Medicine, Yangsan 626-870, Korea; solarhy77@naver.com (H.-Y.K.); yeseonlim@pusan.ac.kr (Y.S.L.); anatomy2017@pusan.ac.kr (S.Y.H.); qhrrn79@naver.com (C.L.); osldy@naver.com (D.Y.L.); moon@pusan.ac.kr (Y.M.); gynsong@gmail.com (Y.J.S.); 3Department of Anatomy and Convergence Medical Sciences, Pusan National University College of Medicine, Yangsan 626-870, Korea; 4Department of Convergence Medical Sciences, Pusan National University College of Medicine, Yangsan 626-870, Korea; 5Department of Obstetrics and Gynecology, Pusan National University College of Medicine, Yangsan 626-870, Korea

**Keywords:** cisplatin, fish collagen peptides, thymic epithelial cells, reactive oxygen species, apoptosis, MAPK (p38 MAPK, JNK, and ERK) pathway

## Abstract

Thymic epithelial cells (TECs) account for the most abundant and dominant stromal component of the thymus, where T cells mature. Oxidative- or cytotoxic-stress associated injury in TECs, a significant and common problem in many clinical settings, may cause a compromised thymopoietic capacity of TECs, resulting in clinically significant immune deficiency disorders or impairment in the adaptive immune response in the body. The present study demonstrated that fish collagen peptides (FCP) increase cell viability, reduce intracellular levels of reactive oxygen species (ROS), and impede apoptosis by repressing the expression of Bax and Bad and the release of cytochrome c, and by upregulating the expression of Bcl-2 and Bcl-xL in cisplatin-treated TECs. These inhibitory effects of FCP on TEC damage occur via the suppression of ROS generation and MAPK (p38 MAPK, JNK, and ERK) activity. Taken together, our data suggest that FCP can be used as a promising protective agent against cytotoxic insults- or ROS-mediated TEC injury. Furthermore, our findings provide new insights into a therapeutic approach for the future application of FCP in the prevention and treatment of various types of oxidative- or cytotoxic stress-related cell injury in TECs as well as age-related or acute thymus involution.

## 1. Introduction

The thymus is the primary lymphoid organ composed of multiple cell types creating a unique microenvironment for producing immunocompetent T cells from bone marrow-derived T cell progenitors/precursors, thereby playing a crucial role in the development of the host adaptive immune system. T cells are not only trained to recognize and eliminate foreign antigens during development in the thymus, but also to tolerate self-antigens through positive and negative selection. This thymic education of T cells is mainly orchestrated by thymic epithelial cells (TECs) that play a pivotal role in the multistep processes, including the homing and clonal expansion, survival, and maturation of the immature T cells [1].

The thymus is a peculiar organ in our body that undergoes progressive involution or atrophy with age. It is called age-related or physiological thymic involution (or atrophy), resulting in a deterioration of its T cell generation ability. In addition, many stimuli including infection, radiation, stress, toxic substances, pregnancy, steroid hormones, malnutrition, immunosuppressive drugs, such as cyclosporine and dexamethasone, chemotherapeutic agents, such as cyclophosphamide, cisplatin, methotrexate, and taxanes, dangerous substances, and harmful biological processes can cause a condition known as acute or accidental thymic involution (or atrophy) [2,3]. Regardless of the type and causative agents of thymic involution, it is usually connected with primary TEC injury [4,5,6], ultimately leading to a decreased thymus production and outward migration of mature T cells, and increased severity and susceptibility to infections, cancer, and autoimmune diseases [7,8,9].

Oxidative stress is a disturbance in the balance between reactive oxygen species (ROS) activities and antioxidant defense mechanisms associated with detoxification of the harmful effects of ROS, which affects the induction of thymic involution [10]. Many endogenous and exogenous stimuli trigger ROS production, damaging DNA, cellular proteins/lipids, and cell membrane, and inducing an inflammatory response, leading to apoptotic or necrotic cell death [11].

Chemotherapies with platinum-based anticancer drugs are standard treatments for various cancers, including ovarian cancer, cervical cancer, lung cancer, head and neck cancer, bladder cancer, and lymphoma [12,13]. Cisplatin was the first widely used platinum-based chemotherapy drug and has been the basis agent for treating a broad spectrum of cancers [14]. However, despite its huge potential as a chemotherapy regimen, cisplatin has been linked to several toxic side effects, including nephrotoxicity, lymphosuppression, myelosuppression, ototoxicity, cardiotoxicity, hepatotoxicity, and neurotoxicity, through various mechanisms, such as oxidative stress, apoptosis, inflammation, and autophagy [15,16]. Cisplatin mainly displays its cytotoxic activity by tipping the redox scale favoring oxidative stress, leading to mitochondrial membrane permeabilization and DNA damage [17,18].

Currently, fish collagen peptides (FCP) are gaining increasing attention due to their purported safety [19,20,21] and diverse biological activities, such as antioxidant activity [22,23], neuroprotective effects [24], anti-aging effects [25], and wound healing effects [26]. Furthermore, oral administration of FCP was reported to diminish the production of pro-inflammatory cytokines, such as tumor necrosis factor-α (TNF-α) and nitric oxide (NO) in rat synoviocytes, and NO and C-reactive protein in diabetic patients with chronic inflammation [27,28]. However, there are still many questions left unanswered about the efficacy of FCP in protecting TEC injury.

Here, the present study shows that FCP protects TECs against cisplatin-induced cytotoxic and oxidative injury by inhibiting MAPK signal transduction pathways.

## 2. Results

### 2.1. FCP Promote Cell Proliferation and Inhibit Cisplatin-Induced Cytotoxicity

A WST-1-based colorimetric cell proliferation assay was used evaluate the ability of FCP to facilitate cell proliferation. Treatment of TECs with FCP for 24 h significantly enhanced cell proliferation at concentrations of 0.01, 0.05, 0.08, 0.1 and 0.15%, by 19.6% (*p* < 0.01), 23.7% (*p* < 0.001), 27.1% (*p* < 0.001), 19.3% (*p* < 0.01), and 11.2% (*p* < 0.05), respectively, compared with the control (Figure 1). At 48 h, the rates of cell proliferation in the FCP-treated group versus the control at concentrations of 0.01%, 0.05%, 0.08%, 0.1%, 0.15%, and 0.2% were greatly increased by 30.9% (*p* < 0.001), 44.8% (*p* < 0.001), 56% (*p* < 0.001), 38.6% (*p* < 0.001), 21.7% (*p* < 0.01), and 12.8% (*p* < 0.01), respectively, compared with the control (Figure 1).

Cellular cytotoxicity and morphology were assessed to examine the level of cisplatin-induced cell injury. Here, the WST-1-based colorimetric cell viability assay after treatment with cisplatin for 24 h at concentrations of 5, 10, and 20 μM revealed a significant decrease in cell number by 26.7% (*p* < 0.01), 36.9% (*p* < 0.01), and 57.9% (*p* < 0.001), respectively, relative to the control (Figure 2). Cisplatin treatment for 48 h at concentrations of 5, 10, and 20 μM strongly reduced cell viability by 61.3% (*p* < 0.001), 87.9% (*p* < 0.001), and 100% (*p* < 0.001), respectively, compared with the control (Figure 2). These cytotoxicity results were also concurrent with the morphological changes observed by phase contrast microscopy (Figure 2), revealing dose- and time-dependent cytotoxic effects of cisplatin in TECs.

Subsequently, the protective effect of FCP was investigated on cisplatin-induced cytotoxicity in TECs by WST-1 using the phase contrast microscopic assays. As shown in Figure 3, exposure of TECs to cisplatin at concentrations of 5, 7.5, 10 and 15 μM for 24 h led to a reduced cell viability, by 22.5% (*p* < 0.01), 30.3% (*p* < 0.001), 43.1% (*p* < 0.001), and 58.7% (*p* < 0.001), respectively, compared with the control. However, 0.08% FCP pretreatment prior to cisplatin treatment (5, 7.5, 10, and 15 μM for 24 h) resulted in the enhancement of cellular viability, by 41.1% (*p* < 0.001), 42.5% (*p* < 0.01), 36.7% (*p* < 0.01), and 24.04% (*p* < 0.001), respectively, compared with the cisplatin alone treatment group. This result, therefore, indicates the protective role of FCP against cisplatin-induced TEC damage (Figure 3).

Furthermore, molecular mechanisms underlying the protective effect of FCP on TECs injured by cisplatin were explored by analyzing the expression of apoptosis- and cell cycle-related proteins. Cisplatin treatment significantly reduced the expression of anti-apoptotic molecules, Bcl-2 and Bcl-xL, by 24.4% (*p* < 0.001) and 21.2% (*p* < 0.001), respectively, and enhanced the expression of pro-apoptotic molecules, Bax, Bad and cytochrome-c, by 40.1% (*p* < 0.001), 20.8% (*p* < 0.001), and 40.5% (*p* < 0.01), respectively (Figure 4A), whereas it significantly suppressed the expression of the key cell cycle regulatory molecules, cyclin D1 and CDK1 proteins by 52.1% (*p* < 0.001) and 35.5% (*p* < 0.001), respectively, compared with the untreated control (Figure 4B). Notably, all these cisplatin-induced alterations in the expression of apoptosis- and cell cycle-related proteins almost returned to their normal levels after 0.08% FCP pretreatment for 24 h. The cisplatin-induced downregulated expression of Bcl-2, Bcl-xL, cyclin D1, and CDK1 proteins increased due to FCP by 26% (*p* < 0.001), 22.6% (*p* < 0.01), 29.9% (*p* < 0.05), and 42.8% (*p* < 0.001), respectively, (Figure 4A,B), whereas the cisplatin-induced upregulated expression of Bax, Bad and cytochrome-c was reversed following exposure to FCP by 58% (*p* < 0.001), 39.3% (*p* < 0.001), and 29% (*p* < 0.001), respectively, compared with the cisplatin alone treatment group (Figure 4A).

### 2.2. FCP Attenuate Cisplatin-Induced ROS Generation

To measure changes in the cellular levels of ROS in response to cisplatin with or without FCP pretreatment, TECs were pretreated with 0.08% FCP for 24 h followed by cisplatin exposure at 10 µM for 24 h (Figure 5A,B). Treatment with cisplatin significantly increased the ROS level as detected by fluorescence microscopy using the oxidant-sensing fluorescent probe 2′,7′-dichlorodihydrofluorescein diacetate (DCFH-DA) by 34.4% (*p* < 0.05) compared with the control (Figure 5A,B).

Pretreatment with FCP showed a significant decrease in the cisplatin-induced ROS release by 72.1% (*p* < 0.01) compared with the cisplatin alone-treated group, indicating its restorative effect on the endogenous antioxidant defense mechanism impaired by cisplatin (Figure 5A,B). Furthermore, treatment with N-acetyl cysteine (NAC), a common ROS scavenger, at a concentration of 5 mM for 2 h, significantly reduced the cisplatin-enhanced ROS level in TECs, by 69.2% (*p* < 0.001) compared with the cisplatin alone-treated group (Figure 5A,B). These findings also indicate that the inhibitory action of FCP on cisplatin-induced cytotoxicity in TECs is attributed to its antioxidant effect.

### 2.3. FCP Alleviate Cisplatin-Induced TEC Cytotoxicity through Suppression of MAPK Pathway

MAPK cascades are key signaling pathways that regulate various cellular processes, including stress and inflammatory responses, as well as cell proliferation, differentiation, apoptosis, and motility under both normal and pathological conditions [29,30]. Since cisplatin-induced cell death has been shown to be dependent on the MAPK pathways in several cell types [31], we first tested if cisplatin could activate this pathway in TECs. The results of this study revealed an increase in the expression of p-p38 MAPK, p-JNK, and p-ERK by 57.9% (*p* < 0.001), 16.8% (*p* < 0.001), and 21.8% (*p* < 0.001), in the cisplatin alone-treated group, respectively, than in the control, whereas the expression of total p38 MAPK, JNK, and ERK levels remained unaltered, indicating the activation of p38 MAPK, JNK, and ERK pathways after cisplatin treatment in TECs (Figure 6). However, these elevated levels of p-p38 MAPK, p-JNK and p-ERK were attenuated by the pretreatment with 0.08% FCP for 24 h, by 37.3% (*p* < 0.01), 15.3% (*p* < 0.001), and 58.5% (*p* < 0.001), respectively, compared with the cisplatin alone-treated group (Figure 6).

To further explore the role of MAPK pathways in cisplatin-induced cell cytotoxicity, the cells were treated with a selective p38 MAPK inhibitor (SB203580), JNK inhibitor (SP600125), or ERK inhibitor (U0126). Analysis of the cell viability by WST-1 assay after TECs were treated with 0.08% FCP, 10 µM SB203580, SP600125, and U0126 for 24 h, followed by treatment with or without 10 µM cisplatin for 24 h, showed that FCP pretreatment caused a significant reduction of cisplatin-induced cytotoxicity, similar to all MAPK inhibitor pretreatments, whereas SB203580, SP600125 or U0126 alone did not affect cell viability (Figure 7).

Confirmation of the effect of FCP treatment on the cisplatin-induced expression of MAPK was performed using western blot analysis after TECs were treated with 0.08% FCP and 10 µM SB203580, SP600125, and U0126 for 24 h, followed by treatment with or without 10 µM cisplatin for 24 h. Notably, we observed that FCP pretreatment potently repressed the cisplatin-induced upregulated expression of p-p38 MAPK, p-JNK, and p-ERK, as was the case with MAPK inhibitor pretreatment (Figure 8A–C).

Together, these results indicate that FCP acts as a potent inhibitor of MAPK pathways, which exerts a protective effect against cisplatin-induced cytotoxicity via suppression of p38 MAPK, JNK, and ERK pathway in TECs.

### 2.4. FCP Prevent Cisplatin-Induced ROS Generation by Inhibition of MAPK Signaling

To investigate whether p38 MAPK, JNK and ERK activation is associated with cisplatin-induced oxidative cell injury in TECs, we examined the effect of p38 MAPK, JNK and ERK inhibition on the cisplatin-induced ROS production. We also assessed the protective effect of FCP on cisplatin-induced ROS release in TECs using the DCFH-DA assay. As shown in Figure 9A–C, the exposure of TECs to 10 μM cisplatin for 24 h caused an increase in ROS level, by 29.5% (*p* < 0.01), 34% (*p* < 0.001) and 33.8% (*p* < 0.001) *versus* the control. However, the enhanced level of ROS induced by cisplatin treatment was significantly declined by the treatment with 10 μM SB203580, 10 μM SP600125 or 10 μM U0126 for 24 h by 25.2% (*p* < 0.01), 46.3% (*p* < 0.01), and 29.6% (*p* < 0.05), respectively, and with 0.08% FCP for 24 h by 37.8% (*p* < 0.01), 79% (*p* < 0.001) and 50.3% (*p* < 0.05), respectively (Figure 9A–C).

As shown in Figure 10, the treatment of TECs with 10 μM cisplatin for 24 h resulted in the marked upregulation of p-p38 MAPK, p-JNK, and p-ERK expression compared with the control group by 72.1% (*p* < 0.001), 28.8% (*p* < 0.05), and 56.6% (*p* < 0.001), respectively, while the amount of total p38 MAPK, p-JNK, and p-ERK levels were unaltered. To decipher the role of p38 MAPK, JNK, and ERK in cisplatin-induced oxidative cellular injury in TECs, we investigated the effect of NAC on the cisplatin-induced expression of p-p38 MAPK, p-JNK, and p-ERK in TECs. Pretreatment of cells with FCP for 24 h or NAC for 2 h prior to cisplatin treatment completely abolished the cisplatin-induced phosphorylation of p38 MAPK, JNK, and ERK (Figure 10). These findings indicate that p38 MAPK, JNK and ERK activation is important in cisplatin-induced cellular oxidative stress, and FCP exhibits a strong protective effect against cisplatin-induced oxidative cell injury via the suppression of p38 MAPK, JNK, and ERK pathways in TECs.

## 3. Discussion

The results of the present study indicated that FCP derived from tilapia scales exhibited antioxidant and cytoprotective effects on mouse TECs against oxidative stress provoked by cisplatin exposure. In addition, our findings showed increased cellular viability accompanied by FCP-mediated antioxidant activity based on a diminished level of ROS in the cytosol. Furthermore, this study also provides the first molecular evidence for elucidating the function of FCP serving as a cytoprotective agent against TEC damage by cisplatin. Thus, it is proposed that FCP may offer protective effects on TECs against cytotoxic and oxidative stress-induced cellular injury caused by various types of noxious stimuli.

Cisplatin is a highly reactive molecule that exerts its cytotoxic effects mainly through the formation of covalent DNA adducts [32]. In addition, it stimulates the production of intracellular ROS in several types of cells, including hepatocytes [33], pulmonary alveolar cells [34], renal proximal tubule epithelial cells [35], and intestinal epithelial cells [36]. Our findings are consistent with previous studies showing that treatment with antioxidants alleviates the toxic effects of cisplatin, indicating an essential role of oxidative stress in the pathogenesis of cisplatin-induced cell injury in several different types of organs [33,34].

Marine organisms are important sources of bioactive compounds with potential therapeutic applications. In particular, fish collagen-derived peptides are of considerable interest and have drawn great attention recently due to their bioactive functions [37]. Fish collagen has been shown to exhibit microbicidal, anti-inflammatory and anti-skin-aging activities, as well as wound healing and tissue regeneration [37,38,39,40,41]. Despite much having been learnt about the diverse bioactivities of FCP on multiple cell types [42,43]; there is a paucity of information on the biological effects of FCP on TECs. Antioxidant properties of peptides from the diverse sources of fish collagen, such as skin from cod, hoki, and pollock have been demonstrated in many different cell types, such as liver cells [44,45], fibroblasts [46], macrophages [47,48], and keratinocytes [49]. In addition, it was also revealed that FCP enhances the viability of human lung fibroblasts damaged by oxygen radicals [46]. In accordance with these results, the present study showed that FCP from tilapia scales has potent antioxidative and cytoprotective effects on TECs.

Antioxidants, widely used as ingredients in dietary supplements to improve health in sectors of the food and beverage manufacturing industry, have been studied for their potential in the prevention or treatment of several human diseases, such as cardiovascular diseases, diabetes, metabolic syndrome, neurodegenerative disorders, cancer and age-related diseases [50,51]. In addition, they are also used as food preservatives for preventing lipid oxidation. Although synthetic antioxidants, such as butylated hydroxytoluene (BHT) and butylated hydroxyanisole (BHA) have been extensively used due to their high stability, low costs, and wide availability, health risks including carcinogenicity, are of great concern [52]. Thus, there is a growing trend toward replacing synthetic antioxidants with natural antioxidants in the food processing industry [53,54].

It is well documented that ROS generated endogenously or in response to environmental stress have long been implicated in cellular injury, which causes cell death, especially triggered by the dysregulation of the pro- or anti-apoptotic pathways, and tissue damage leading to the development of many diseases [55]. The present study demonstrated that FCP acts as a potent suppressor of TEC apoptosis induced by cisplatin treatment by promoting Bcl-2 and Bcl-xL expression and inhibiting Bad and Bax expression and cytochrome-c release. Taken together, these findings indicate that the amelioration of cisplatin-induced cytotoxicity by FCP in TECs is mediated by their antioxidant and anti-apoptotic properties. The discovery of the protective mechanisms of FCP for repairing cellular injuries induced by oxidative stress and activation of apoptotic cell death pathway in human TECs would advocate the use of FCP for the prevention and treatment of many clinical conditions linked to excessive ROS generation and perturbation in the apoptotic balance in TECs. This is particularly important because oxidative- or cytotoxic stress-mediated injury in TECs can be a significant problem in many clinical settings, and linked to the induction of acute thymic involution, that may cause compromised thymopoietic capacity in TECs, leading to a severe and clinically significant immune deficiency disorder or dysfunction of the adaptive immunity, and causing the body to be unable to generate appropriate immune responses against invading pathogens.

The present study also demonstrated that FCP can promote the proliferation of TECs. In agreement with our study, Liu and Sun [56], also observed the growth-promoting effect of tilapia FCP on rat bone marrow mesenchymal stem cells. In addition, our previous study suggested that nanofibrous scaffolds containing tilapia FCP contribute to the enhancement of mouse TEC proliferation [57]. Furthermore, Liu et al. [58] showed that bovine collagen peptide compounds promote the proliferation and differentiation of MC3T3-E1 preosteoblasts. These investigations, therefore, corroborate that tilapia FCP exhibit significant growth promotion properties in several types of cells.

MAPK signal transduction pathways are involved in the regulation of a wide variety of fundamental cellular processes, such as cell growth, differentiation, survival, apoptosis, migration, inflammation, and environmental stress responses [59]. To determine the role of the p38 MAPK, JNK, and ERK in cisplatin-induced ROS production and the signaling pathway in TECs, the expression levels of p38 MAPK, JNK, and ERK were analyzed by DCFH-DA, cell proliferation, and western blot assays after treatment with NAC, SB203580, SP600125, U0126, and FCP. Consequently, the cisplatin-elicited p38 MAPK, JNK, and ERK activation was abolished by SB203580, SP600125, and U0126 as well as FCP and NAC, suggesting that cisplatin-induced oxidative stress injury in TECs is mediated by p38 MAPK, JNK and ERK and that FCP, similarly to NAC, notably ameliorates cisplatin-induced oxidative stress in TECs by blocking p38 MAPK, JNK, and ERK activation. In addition, the cisplatin-induced cytotoxic responses were also significantly blocked by SB203580, SP600125, and U0126 as well as FCP. Taken together, these data indicate that FCP plays a critical role in protecting various cytotoxic and oxidative stresses in TECs by repressing the activation of MAPK signal transduction pathways.

## 4. Materials and Methods

### 4.1. Cell Culture and Reagents

Mouse thymic cortical epithelial reticular cells (1308.1) were provided by Dr. Barbara B. Knowles (The Jackson Laboratory, Bar Harbor, ME, USA). The cells were cultured in Dulbecco’s modified Eagle’s medium (DMEM; Hyclone, GE Healthcare Life Sciences, Logan, UT, USA) supplemented with 10% fetal bovine serum (FBS), 100 IU mL^−1^ penicillin, and 100 mg mL^−1^ streptomycin (all from Gibco, Thermo Fisher Scientific, Waltham, MA, USA) in a humidified atmosphere of 5% CO_2_ at 37 °C. Subconfluent cells were harvested with trypsin-EDTA and used for further experiments. Media were replaced every second day.

Cisplatin, 2′,7′-dichlorodihydrofluorescein diacetate (DCFH-DA), N-acetyl-L-cysteine (NAC), 4′,6-diamidino-2-phenylindole (DAPI), and bicinchoninic acid (BCA) were obtained from Sigma-Aldrich (St. Louis, MO, USA). Antibodies against ERK, phospho-ERK (*p*-ERK), JNK, phospho-JNK (p-JNK), p38 MAPK, phospho-p38 MAPK (p-p38 MAPK), cytochrome-c, and cyclin D1 were supplied by Cell Signaling Technology (Cambridge, MA, USA). The antibodies against Bcl-2, Bcl-xL, Bax, Bad, and CDK1 were obtained from Abcam (Cambridge, UK). Additionally, an antibody against β-actin was bought from Santa Cruz Biotechnology (Santa Cruz, CA, USA). The p-p38 MAPK inhibitor (SB203580), p-JNK/MAPK inhibitor (SP600125), and p-ERK/MAPK inhibitor (U0126) were purchased by Tocris Bioscience (Ellisville, MO, USA). FCP extracted from tilapia were provided by Geltech (Busan, Korea), and their physicochemical properties were described in our previous study [49]. All other reagents and compounds used were supplied from Sigma-Aldrich.

### 4.2. Cell Viability Assay

After TECs (8 × 10^3^ cells/well) in 96-well flat-bottom culture plates (SPL Life Sciences, Pocheon, Korea) were treated with the indicated doses of FCP for 24 h with or without cisplatin. The cell viability was determined using the colorimetric WST-1 conversion assay (EZ-Cytox assay kit, Daeil Lab Service, Seoul, Korea). A WST-1 reagent (total 10 μL) was added to each well, and cells were incubated for 2 h in a humidified incubator at 37 °C under 5% CO_2_. The absorbance of the formazan dye, generated by the reaction of dehydrogenase with WST-1 in the metabolically active cells, was measured using a microplate reader (Tecan, Männedorf, Switzerland) at 450 nm according to the manufacturer’s instructions, and the percent cell viability was calculated. The experiments were performed in triplicate.

### 4.3. Measurement of ROS

The effect of FCP on the cisplatin-induced generation of ROS in TECs was detected by DCFH-DA, a ROS-sensitive fluorescent probe, under a fluorescent microscope. Cell-permeable DCFH-DA is non-fluorescent, but in the presence of ROS, when this dye is oxidized, it is converted to a highly fluorescent 2′,7′-dichlorofluorescein (DCF) [60]. TECs (1 × 10^5^ cells/well) in 6-well culture plates were treated with 0.08% FCP for 24 h before treatment with cisplatin (10 µM) for 24 h. After removing the medium from wells, the cells were washed with phosphate buffered saline and then incubated with 10 μM DCFH-DA in fresh serum-free medium for 30 to 40 min in a humidified incubator at 37 °C with 5% CO_2_ under dark conditions. The labeled cells were observed with an epi-fluorescence microscope (BX50, Olympus, Tokyo, Japan). Photomicrographs were acquired digitally at 1360 × 1024 pixel resolution with an Olympus DP70 digital camera. Furthermore, the DCF fluorescence was measured using a fluorescent microplate reader (SpectraMax M2e, BioTek, Winooski, VT, USA) at 495–529 nm. To minimize the possible photo-oxidation of the probe and or photo-reduction of DCF, the plates were covered with aluminum foil to shield the probe from light.

### 4.4. Western Blot Analysis

To determine protein expression levels, TECs (8 × 10^5^ cells/dish), after reaching 70–80% confluency in 60 mm culture dishes (SPL Life sciences), were treated with 0.08% FCP for 24 h before treatment with cisplatin (10 µM) for 24 h. Cells from each set of experiments were harvested and washed twice in cold Tris-buffered saline (TBS, 20 mM Tris-HCl, 150 mM NaCl, pH 7.4). For the western blot analysis, cells were lysed in 100 μL RIPA cell lysis buffer with EDTA (GenDEPOT, Barker, TX, USA) containing a protease inhibitor mixture (Roche, Basel, Switzerland). Samples were kept on ice for 30 min, vortexing briefly (15 s) every 2–3 min. Then, the lysates were centrifuged at 14,000 RPM for 30 min at 4 °C, and the protein concentration was measured using a BCA protein assay (Sigma-Aldrich). Equal amounts of protein samples were heated for 10 min at 70 °C in Bolt LDS sample buffer (Invitrogen, Waltham, MA, USA) and separated by 10% sodium dodecyl sulfate (SDS)-polyacrylamide gel electrophoresis (PAGE, Invitrogen) at 200 V for 25 min, using a Mini-Protean III system (Bio-Rad, Hercules, CA, USA). Proteins were transferred to a polyvinylidene difluoride (PVDF) membrane (GE Healthcare Life science) at 20 V for 1 h. The nonspecific binding was blocked with 3% bovine serum albumin (BSA) in TBS buffer containing 0.1% Tween 20 (TBST buffer), incubated with the indicated primary antibodies at a dilution of 1:500–1:2000 with 5% BSA in TBST overnight at 4 °C with anti-p38 MAPK, anti-p-p38 MAPK, anti-JNK, anti-p-JNK, anti-ERK, anti-p-ERK, anti-Bax, anti-cytochrome-c, anti-Bcl-2, anti-Bcl-xL, anti-Bad, anti-cyclin D1, anti-CDK1, and anti-β-actin (Appendix A).

On the following day, the membrane was washed with TBST buffer thrice and incubated with secondary antibodies, namely, anti-rabbit IgG HRP conjugate (Cell Signaling Technology) and anti-mouse IgG HRP conjugate (Cell Signaling Technology), at a dilution of 1:10,000 with 3% BSA in TBST for 1 h at room temperature. Subsequently, the membrane was washed thrice with TBST. Immunoreactivity was detected with enhanced chemiluminescence (ECL, Super Signal West Pico Chemiluminescent Substrate kit, Pierce, Rockford, IL, USA) according to the manufacturer’s instructions. Images were captured and quantified using a LAS-3000 imaging system (Fujifilm, Tokyo, Japan).

### 4.5. Statistical Analysis

The results of the present study were expressed as the mean ± SD under all conditions. Statistical analysis was performed using a two-tailed Student’s *t*-test. Statistically significant differences were considered at *p* < 0.05.

## 5. Conclusions

The findings of the present study demonstrate for the first time that FCP stimulates proliferation, and ameliorates cisplatin-induced cytotoxicity and oxidative stress in TECs. In addition, it was shown that the inhibitory effects of FCP on cytotoxicity are likely associated with suppression of the ROS and MAPK (p38 MAPK, JNK, and ERK) signaling pathways. Therefore, these results suggest that FCP may be a promising protective agent in TEC injury induced by cytotoxicity and oxidative stress elaborated by chemotherapeutic drugs, such as cisplatin and other cytotoxic agents. Furthermore, the data of the current study may provide new insights into the therapeutic approach for the future application of FCP in the prevention and treatment of a variety of the cytotoxic and oxidative stress-mediated injuries in TECs as well as acute or age-related thymic involution.

## Figures and Tables

**Figure 1 marinedrugs-20-00232-f001:**
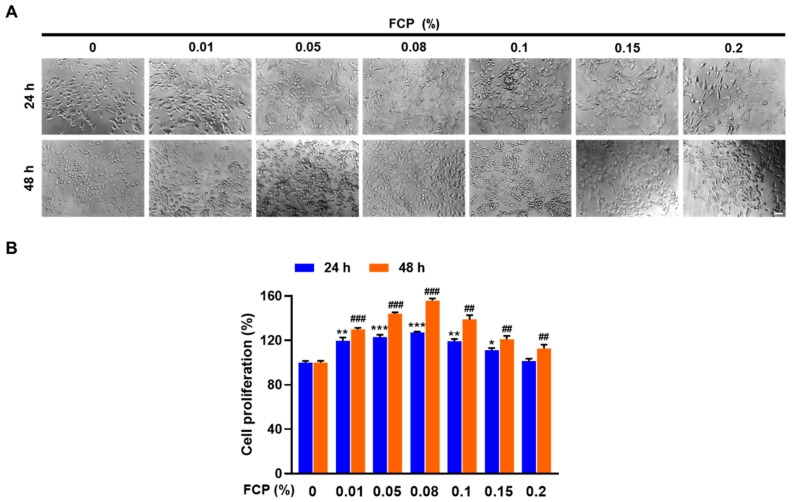
Stimulatory effects of FCP on cell proliferation. Proliferation of TECs was measured by phase contrast microscopy (**A**) and cell viability assay (**B**) as described in Materials and Methods. The treatment of TECs with FCP for 24 and 48 h significantly enhanced cell proliferation. Results are presented as the means ± SD of three independent experiments. * *p* < 0.05, ** *p* < 0.01, *** *p* < 0.001 vs. the control at 24 h; ^##^ *p* < 0.01, ^###^ *p* < 0.001 vs. the control at 48 h. Scale bar = 50 µm.

**Figure 2 marinedrugs-20-00232-f002:**
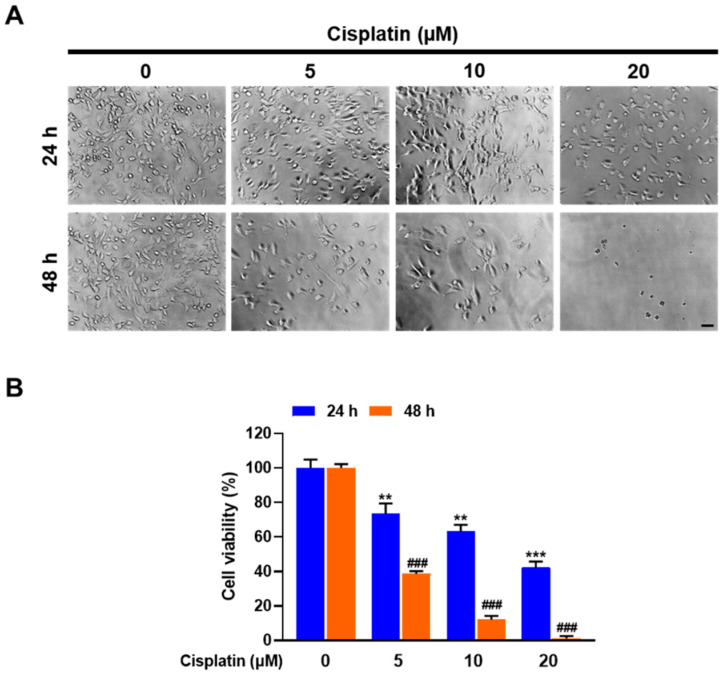
Cytotoxic effects of cisplatin on TECs were measured by phase contrast microscopy (**A**) and cell cytotoxicity assay (**B**) as described in Materials and Methods. The treatment of TECs with 5, 10, and 20 µM cisplatin for 24 and 48 h significantly attenuated cell viability. Results are presented as the means ± SD of three independent experiments. ** *p* < 0.01, *** *p* < 0.001 vs. the control at 24 h; ^###^ *p* < 0.001 vs. the control at 48 h. Scale bar = 50 µm.

**Figure 3 marinedrugs-20-00232-f003:**
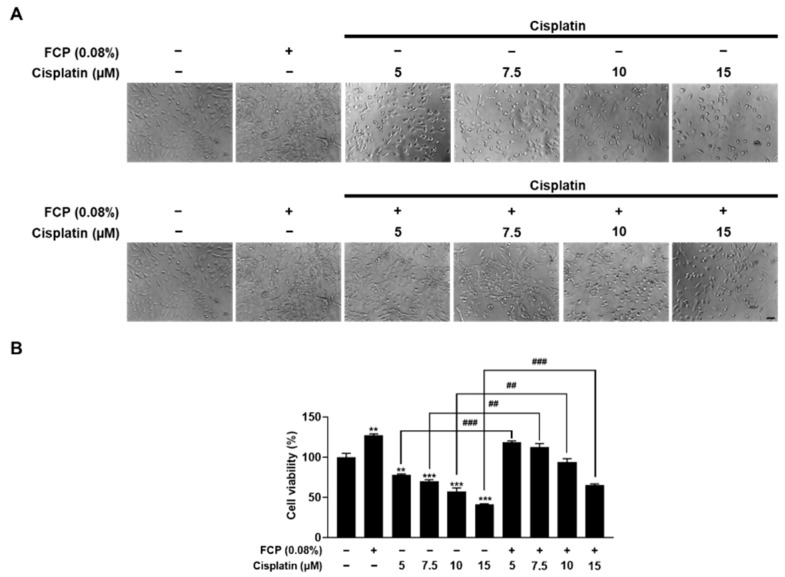
Protective effect of FCP on cisplatin-induced cytotoxicity in TECs. Cell viability was measured by phase contrast microscopy (**A**) and WST-1 assay (**B**) in TECs as described in Materials and Methods. The decreased cell number induced by cisplatin treatment (5, 7.5, 10, and 15 µM) was significantly restored by treatment with 0.08% FCP for 24 h. Results are presented as the means ± SD of three independent experiments. ** *p* < 0.01, *** *p* < 0.001 vs. the control. ^##^ *p* < 0.01, ^###^ *p* < 0.001 vs. the cisplatin alone-treated group. Scale bar = 50 µm.

**Figure 4 marinedrugs-20-00232-f004:**
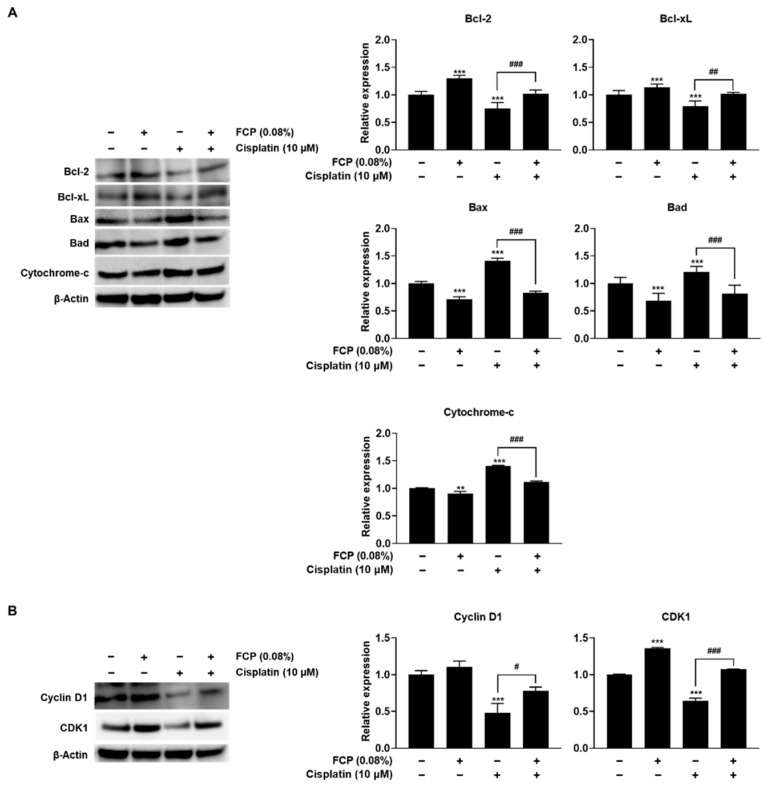
Western blot analysis on the inhibitory effects of FCP on cisplatin-induced altered expression of apoptosis- and proliferation-related proteins in TECs. FCP pretreatment in cisplatin alone-treated TECs significantly enhanced the levels of Bcl-2 and Bcl-xL, reduced the levels of Bad, Bax, and cytochrome-c (**A**), and elevated the levels of cyclin D1 and CDK1 compar with the cisplatin alone-treated group (**B**). Results are presented as the means ± SD of three independent experiments. ** *p* < 0.01, *** *p* < 0.001 vs. the control. ^#^ *p* < 0.05, ^##^ *p* < 0.01, ^###^ *p* < 0.001 vs. the cisplatin alone-treated group.

**Figure 5 marinedrugs-20-00232-f005:**
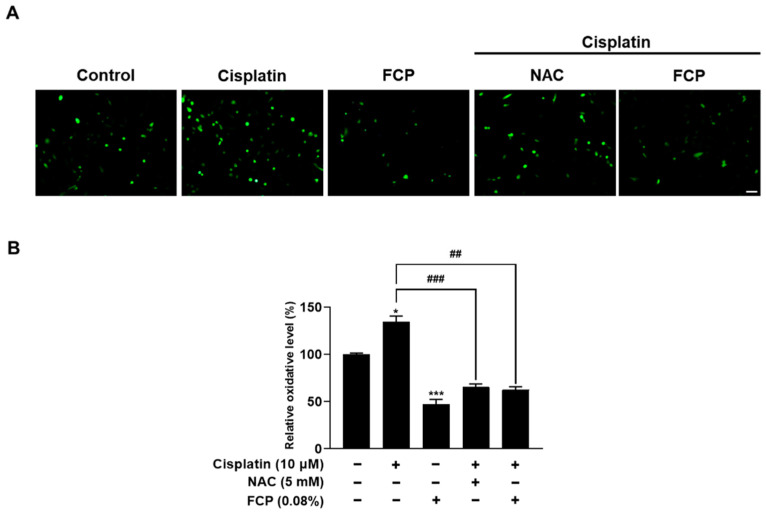
Inhibitory effects of FCP and NAC on cisplatin-induced ROS generation in TECs. Intracellular ROS levels were determined via fluorescence microscopy (**A**) and spectroscopy (**B**) using DCFH-DA in TECs. The increased ROS level induced by 10 µM cisplatin treatment was significantly attenuated by pretreatment with 0.08% FCP for 24 h and NAC for 2 h. Quantification of staining intensity was measured by ImageJ software. Results are presented as the means ± SD of three independent experiments. * *p* < 0.05, *** *p* < 0.001 vs. the control. ^##^ *p* < 0.01, ^###^ *p* < 0.001 vs. the cisplatin alone-treated group. Scale bar = 50 µm.

**Figure 6 marinedrugs-20-00232-f006:**
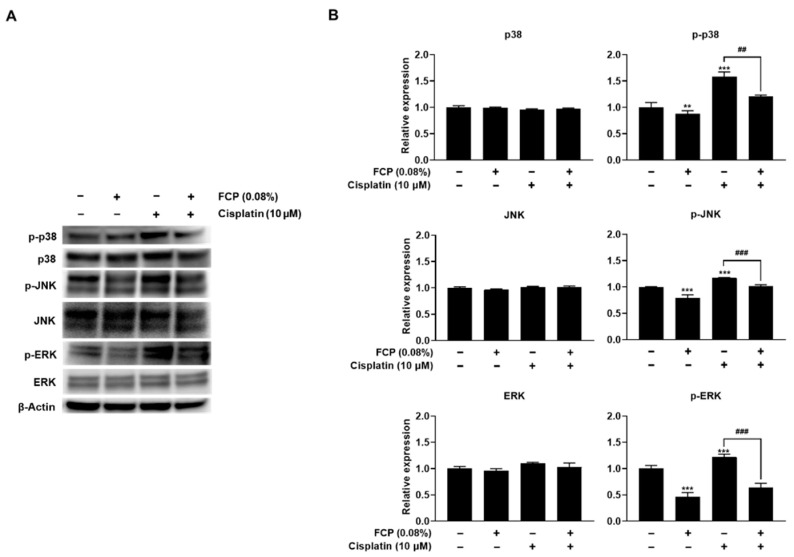
Inhibitory effects of FCP on cisplatin-induced activation of p38 MAPK, JNK, and ERK signaling pathway. The expression of p38 MAPK, JNK, and ERK was increased in the cisplatin alone-treated TECs, as assessed by Western blot analysis (**A**). The pretreatment of TECs with FCP blocked the cisplatin-induced phosphorylation of p38 MAPK, JNK, and ERK. Bar graphs depict relative densitometry quantitation of each protein normalized to β-actin (**B**). Results are presented as the means ± SD of three independent experiments. ** *p* < 0.01, *** *p* < 0.001 vs. the control. ^##^ *p* < 0.01, ^###^ *p* < 0.001 vs. the cisplatin alone-treated group.

**Figure 7 marinedrugs-20-00232-f007:**
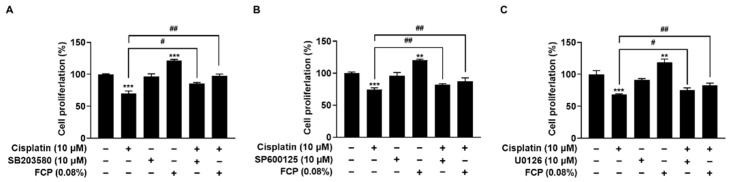
FCP promotes TECs proliferation via activation of p38 MAPK (**A**), JNK (**B**), and ERK (**C**) signaling pathways. The treatment of TECs with FCP for 24 h significantly enhanced cell proliferation. The cisplatin-induced cell cytotoxicity was recovered by treatment with FCP, SB203580, SP600125, and U0126 in TECs. Results are presented as the means ± SD of three independent experiments. ** *p* < 0.01, *** *p* < 0.001 vs. the control at 24 h; ^#^ *p* < 0.05, ^##^ *p* < 0.01, vs. the control at 48 h.

**Figure 8 marinedrugs-20-00232-f008:**
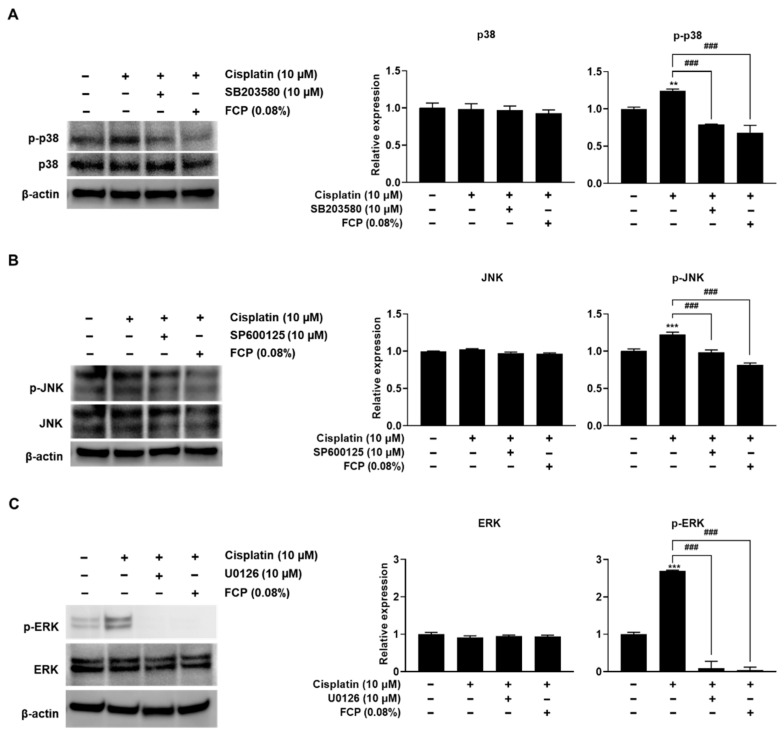
Inhibitory effects of FCP, SB203580 (**A**), SP600125 (**B**), and U0126 (**C**) on cisplatin-induced activation of p38 MAPK, JNK, and ERK signaling pathway in TECs. The expression of p-p38 MAPK, p-JNK, and p-ERK was increased in cisplatin-treated TECs. The pretreatment of TECs with FCP, SB203580 (**A**), SP600125 (**B**), and U0126 (**C**) blocked the cisplatin-induced phosphorylation of p38 MAPK, JNK, and ERK. Results are presented as the means ± SD of three independent experiments. ** *p* < 0.01, *** *p* < 0.001 vs. the control. ^###^ *p* < 0.001 vs. the cisplatin alone-treated group.

**Figure 9 marinedrugs-20-00232-f009:**
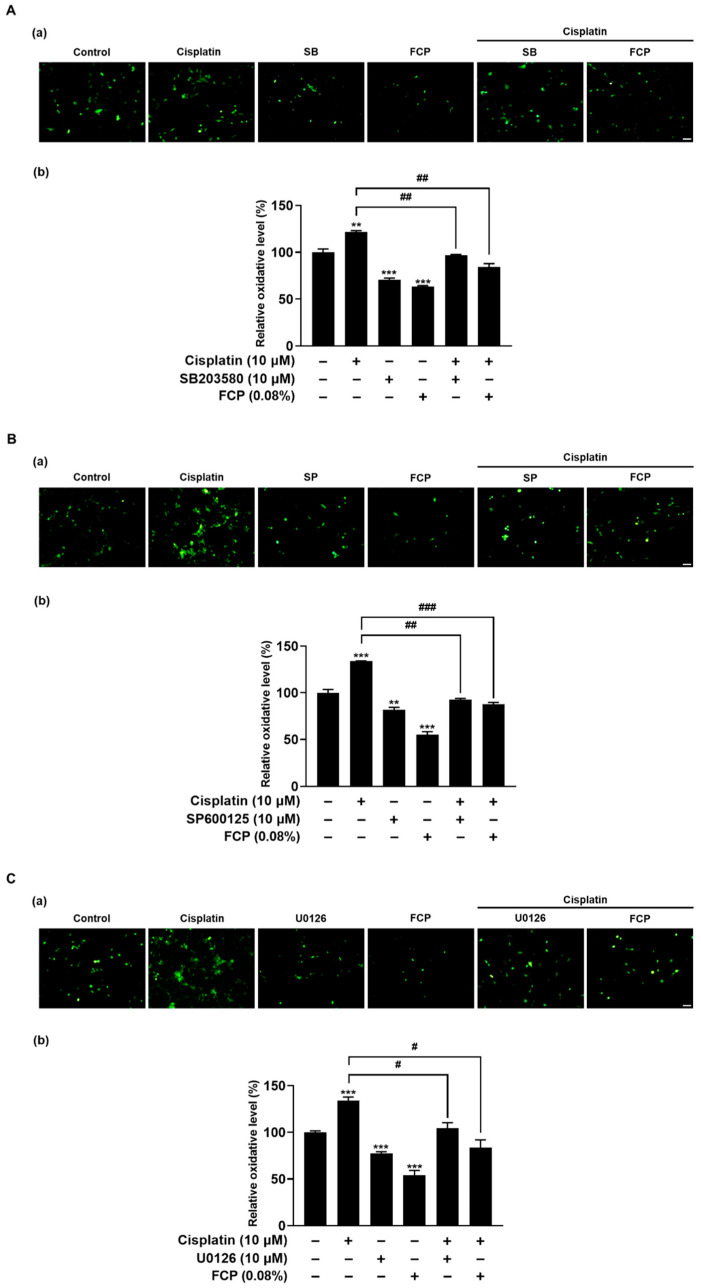
Inhibitory effects of FCP on cisplatin-induced ROS generation in TECs via activation of p38 MAPK, JNK and ERK signaling pathway. Intracellular ROS levels were determined via fluorescence microscopy (a) and spectroscopy (b). The increased ROS production induced by cisplatin returned to the control level by treatment with FCP, SB203580 (**A**), SP600125 (**B**) and U0126 (**C**) in TECs. Results are presented as the means ± SD of three independent experiments. ** *p* < 0.01, *** *p* < 0.001 vs. the control. ^#^ *p* < 0.05, ^##^ *p* < 0.01, ^###^ *p* < 0.001 vs. the cisplatin alone-treated group. Scale bar = 50 µm.

**Figure 10 marinedrugs-20-00232-f010:**
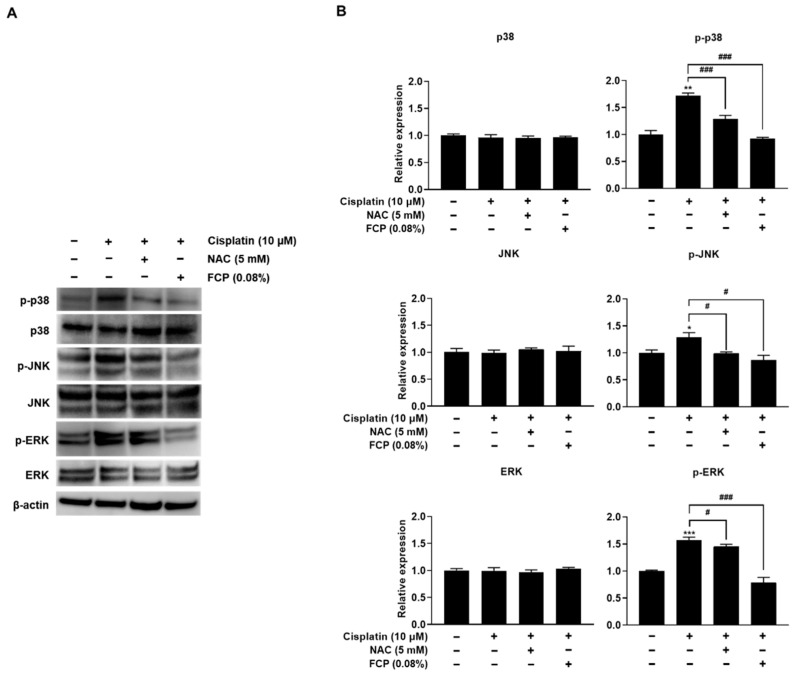
Inhibitory effects of FCP and NAC on cisplatin-induced activation of p38 MAPK, JNK, and ERK signaling pathways in TECs. The expression of p38 MAPK, JNK, and ERK was increased in cisplatin-treated TECs, as assessed by Western blot analysis (**A**). The pretreatment of TECs with FCP or NAC blocked the cisplatin-induced phosphorylation of p38 MAPK, JNK, and ERK. Bar graphs depict relative densitometry quantitation of each protein normalized to β-actin (**B**). Results are presented as the means ± SD of three independent experiments. * *p* < 0.01, ** *p* < 0.01, *** *p* < 0.001 vs. the control. ^#^ *p* < 0.05, ^###^ *p* < 0.001 vs. the cisplatin alone-treated group.

## Data Availability

Not applicable.

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
