# Peer review of "Fish Collagen Peptides Protect against Cisplatin-Induced Cytotoxicity and Oxidative Injury by Inhibiting MAPK Signaling Pathways in Mouse Thymic Epithelial Cells"

_marinedrugs, 2022, doi:10.3390/md20040232_

Round 1

Reviewer 1 Report

It is pleasant reading this article. The authors design a well-constructed hypothesis about Fish Collagen Peptides Protect Against Cisplatin-Induced Cytotoxicity and Oxidative Injury. Besides, I have some concerns and suggestions about the manuscripts.

Title: 1. Words like “Inhibiting p38 MAPK, JNK, and ERK” in the title should modify its looks like a conclusion.

  1. p38 and its phosphor form in figure 8 are not informative, replaced with another replicate.
  2. b-Actin blots are the same in fig A and B, western blot control in figure 4. And the same thing is in figure 10. Why this?

Author Response

We attached the file for a point-by-point response to the reviewer’s comments.

Reviewer 2 Report

Dear Editor,

The manuscript is well written and experiments are well performed. In my opinion, it is interesting, and I recommend its publication in the Journal after the following points:

-Why did the authors use SB203580, SP600125, U0126, and NAC for treatments?

-Add references for the material and method section.

-How did the authors select the protective dose of FCP and NAC, …?

-Provide the graphical abstract.

-How did the authors quantify the western results in fig 10? Explain in the method section.

-The discussion part can be improved by providing a more critical discussion of recent related literature.

Author Response

(The authors gave the same response as above.)

Round 2

Reviewer 2 Report

The authors have responded adequately to my previous comments. I have no more comments.